# The Crucial Role of Breeder and Dog Owners Associations in Safeguarding Genetic Heritage of Endangered Balearic Dog Breeds: Gender Preference and Registry Adscription

**DOI:** 10.3390/ani14040639

**Published:** 2024-02-16

**Authors:** José Manuel Alanzor Puente, Águeda Laura Pons Barro, Antonio González Ariza, Carmen Marín Navas, Juan Vicente Delgado Bermejo, Francisco Javier Navas González

**Affiliations:** 1Institut de Reserca i Formaciò Agroalimentaria de les Illes Balears, Conselleria d’Agricultura, Pesca i Alimentació, Majorca, Govern Illes Balears, 07009 Palma, Spain; jalanzor@irfap.es (J.M.A.P.); apons@irfap.es (Á.L.P.B.); 2Centro Agropecuario Provincial de Córdoba, Diputación Provincial de Córdoba, 14014 Córdoba, Spain; aga07@dipucordoba.es; 3Department of Genetics, Faculty of Veterinary Sciences, University of Córdoba, 14071 Córdoba, Spain; v32manac@uco.es (C.M.N.); id1debej@uco.es (J.V.D.B.)

**Keywords:** breeder associations, dog owner involvement, genetic heritage preservation, canonical correlation analysis, breed registries, pedigree awareness, gender preferences

## Abstract

**Simple Summary:**

This study unravels the intricate dance between indigenous dog breeds in the Balearic Isles and their human companions, focusing on breeders and owners. Using a tool called Canonical Correlation Analysis, the researchers explored various factors like breed registries and the number of breeders/owners, revealing interesting patterns. For example, when more female dogs were added to auxiliary registries, there was a drop in foundational registrations, indicating changes in how breeds are documented. A similar pattern emerged with definitive female registrations, hinting at a growing awareness of pedigrees over time. Beyond these records, the study shows that an increase in breeders significantly influences initial registrations, total definitive counts, and overall numbers, showcasing their pivotal role in the early stages of a breed. The study also touches on gender preferences in registrations, noting a historical preference for female entries during foundational stages that shifts towards males in definitive registrations. In essence, the research stresses the collaborative efforts of breeders, owners, and comprehensive registries in preserving the genetic diversity of Balearic dog breeds. The need for ongoing efforts to fill gaps in genealogical data is highlighted for a more accurate understanding of breed dynamics.

**Abstract:**

This study delves into the complex relationships between indigenous dog breeds in the Balearic Isles and their human counterparts, specifically breeders and owners. Using Canonical Correlation Analysis, the research examines variables such as breed registries and the number of breeders/owners, uncovering significant correlations within registries. For example, an increase in female auxiliary registrations corresponds to a decline in foundational registrations, indicating shifts in breed documentation dynamics. Similarly, a rise in definitive female registrations coincides with a decrease in foundational female registrations, suggesting increased pedigree awareness across generations. Beyond registries, the study explores the correlation between breeders/owners and various initial records, highlighting that a notable increase in breeders positively influences initial registrations, definitive totals, and overall counts, underscoring their crucial role in early breed stages. Gender preferences in registrations are noted, with a historical bias towards female entries during foundational stages gradually shifting in favor of males in definitive registrations. In conclusion, the research underscores the interconnected roles of breeders, owners, and comprehensive registries in preserving genetic diversity among Balearic dog breeds, emphasizing the need for ongoing efforts to address gaps in genealogical data for a more accurate understanding of breed dynamics.

## 1. Introduction

One of the most important protection structures established for the safeguarding of endangered dog breeds is the registration of individuals in studbooks [1]. These studbooks are meticulously maintained records that track the lineage, demographics, and vital information of each individual dog within a managed population.

The efficiency of conservation efforts has deep roots on the intricate relationships between indigenous dog breeds in the Balearic Isles (Ca de Conills, Ca Eivissenc, Ca de Bou, Ca de Bestiar, Ca Rater Mallorquí, and Ca Mè Mallorquí) and their human counterparts, specifically breeders and owners, who act as the primary agents in the protection of this genetic resource as a part of their culture and heritage [2]. Studbooks help trace the lineage of each dog through the study of genealogical relationship comprised in the pedigree or using molecular genetic markers [3]. By documenting parentage, they ensure that breeding pairs are genetically diverse, minimizing the risk of inbreeding [4]. This genetic diversity is crucial for the long-term survival of the species. Studbooks facilitate population management by monitoring births, transfers, deaths, among others. These data allow geneticists to make informed decisions about breeding recommendations and conservation programs [5].

When a dog belonging to an endangered breed is born, a unique registration number is assigned to it. This number is recorded in the studbook, along with details such as the dog’s name, date of pedigree issuance, championship titles obtained, breed, variety, sex, coat color, birthdate, microchip identification code, and ancestral data defining its breeding line, the breeder in which it was born and the caretaker who administratively registers it (owner), among others. Additional data to be found in a studbook may be the reproductive status of the animal, either entire or gelded, and the live status of the animal, which means if it is alive or dead, in previously registered dogs.

The Spanish official registry system for dogs encompasses various categories, each serving a unique purpose in documenting breed lineage and characteristics. As described in Alanzor Puente et al. [1], the Foundational Registry (FR) marks the initial registration phase for a new breed or variety, capturing essential details like origins and lineage to establish the breed’s foundation within the official registry system. While it establishes the base for breed morphological standards, the FR provides limited information compared to more comprehensive registries. The Definitive Registry (DR) represents the pinnacle, housing dogs that meticulously meet all breed standards and undergo thorough evaluation [CC]. These purebred representatives adhere to specific criteria such as appearance, temperament, and genetic heritage, making the DR the most accurate and detailed registry type containing complete lineage details. The Auxiliary Registry (AR) accommodates dogs not fully meeting the DR criteria but contributing valuable genetics to maintain genetic diversity within the breed. Although it supplements birth records, the AR may still lack complete lineage details. The Birth Definitive Registry (BDR) confirms birth details and solidifies lineage records, adding certainty to lineage while potentially lacking comprehensive breed history. The Birth Auxiliary Registry (BAR) meticulously records individual puppy births, essential for accurate lineage tracking despite potentially incomplete lineage information. Lastly, the Merit Registry (MR) celebrates exceptional dogs within a breed, focusing on accomplishments rather than birth or lineage, thus encouraging breed development and excellence recognition. Each registry type plays a vital role in preserving breed standards and genetic diversity within the dog breeding community.

Article 5 of Royal Decree 558/2001 [6], which regulates the official recognition of purebred dog breeders’ organizations or associations in Spain, establishes specific requirements for the registration of dogs in a genealogical book when there is no information available about their ancestors. If parents and grandparents are registered in the studbook, puppies also need to have been identified after birth being subject to verification of compliance with the requirements contained in the aforementioned Royal Decree, the mating and litter birth must have been declared on forms signed by the owners of the animals detailing the total number of puppies born and they should meet the racial standard and minimum requirements of the breed as provided in Royal Decree 558/2001 [6]. The adscription to breed standard is checked at different ages depending on each breeders’ association own criteria and requirements. For instance, small breeds may be evaluated from 6 months, medium breeds from 9 months and large breeds from 12 months. 

Article 7 of the same regulation states that the autonomous communities must communicate to the Ministry of Agriculture, Fisheries, and Food the resolutions of granting and extinguishing the recognition of organizations and associations to make the corresponding annotations and modifications. The general registry of organizations or associations of purebred dog breeders is public and informative and is available to all competent authorities in the field. As Article 7 states, genealogical books of purebred dogs shall consist, at least, of the main section (definitive registry) and annex (auxiliary registry) section. In the main section, purebred dogs may be registered. In the annex section (Auxiliary registry), dogs totally or partially lacking genealogical documentation proving their ancestry, but which, due to their ethnic characteristics, can contribute to the improvement of the breed, may be registered. Genealogical books may also have a register of merits, where those specimens that, being registered in the main section of the book, have passed the aptitude tests established for each breed, demonstrating exceptional qualities, shall be registered. For purebred dogs to be registered in the main section, they must either come from parents and grandparents registered in a genealogical book; they must have been identified after birth, subject to verification of compliance with the requirements contained in this Royal Decree; and the mating and litter birth must have been declared on forms signed by the owners of the animals detailing the total number of puppies born and/or meet the racial standard and minimum requirements of the breed as provided in this Royal Decree [6].

For dogs of a breed to be registered in the annex section (Auxiliary registry), they must either conform to the breed standard; be identified after birth, in accordance with the requirements set forth in this Royal Decree; and/or meet the minimum characteristics in accordance with the requirements set forth in this Royal Decree.

Spanish canine breeds must undergo a breed confirmation process conducted by experts appointed by the recognized breeders’ organizations or associations for the purposes provided for in this Royal Decree, in order to verify their suitability for breeding and the absence of transmissible zootechnical defects. A positive result in the breed confirmation will be necessary to qualify specimens for reproductive purposes and inclusion of their future offspring in the genealogical book of that breed.

For the registration of dogs in a studbook where their ancestors are not listed, the owner must documentarily prove the registration of those ancestors in a genealogical book of a Spanish organization or association officially recognized for keeping genealogical records. In case the ancestors are registered in a foreign genealogical book, the criteria established in the corresponding country or, failing that, internationally recognized criteria shall apply. The required documentation must be issued by the association holding the registry of such ancestors.

Dogs originally registered in an official genealogical book of Spain cannot be registered in other Spanish genealogical books unless their owners have provided the corresponding accreditation of having requested the deregistration from the original registry.

In the broader context of the social sciences, breeder associations play a pivotal role of [7]. These associations, serving as knowledge hubs and community forums, contribute not only to the understanding of genetic diversity but also to the broader socio-cultural landscape. The collaborative efforts of breeders within these associations shape cultural perspectives and societal attitudes towards indigenous dog breeds, presenting a multidimensional approach to breed preservation.

Importantly, this study delves into the broader significance of preserving endangered dog breeds [8]. Beyond the genetic diversity highlighted in the study, these breeds have played crucial roles in human societies since their domestication [9]. Many of these dogs have been integral to various human activities, such as hunting [10], shepherding [11], and companionship [12]. Recognizing the historical and functional roles of these breeds adds a layer of urgency to their conservation [13]. Preserving these breeds is not just about safeguarding genetic diversity but also about ensuring the continuation of functions deeply intertwined with human history and culture.

The present paper contributes to the discourse on the conservation of endangered Balearic dog breeds, bridging genetic and social science perspectives. By highlighting the interconnectedness of breeders, owners, and the broader societal context, this research underscores the need for comprehensive efforts to address gaps in genealogical data. In these regards, the study underscores the significant role of breeders, emphasizing their influence on initial registrations, total definitive counts, and the overall trajectory of a breed in its early stages. Gender preferences in registrations are explored, revealing historical biases that evolve over time, reflecting broader societal changes. For this reason, ultimately, this study aimed to enhance our understanding of the intricate dynamics shaping the preservation of both biological and cultural heritages in the context of endangered dog breeds, emphasizing their historical and functional significance in human societies.

## 2. Materials and Methods

### 2.1. Sample

The populations evaluated include Ca de Conills, Ca Eivissenc, Ca de Bou, Ca de Bestiar, Ca Rater Mallorquí, and Ca Mè Mallorquí, each associated with specific breeder organizations. These associations oversee the management and maintenance of studbooks, which serve as invaluable repositories of breed lineage and history. Studbooks record vital information such as registration numbers, pedigree issuance dates, and ancestral data defining breeding lines. Additionally, they facilitate the tracking of championship titles obtained in dog shows, offering insights into the breed’s performance and recognition.

Ca de Bou and Ca de Bestiar boast significant census figures, with 243 and 507 individuals, respectively, indicating their notable presence within the Spanish dog population, particularly in roles such as guarding and shepherding. On the other hand, breeds like the Ca de Conills, despite having a smaller census of 316, still hold significance, especially in hunting and ratting activities. Moreover, breeds like the Ca Rater Mallorquí, Ca Eivisenc, and Ca Mè Mallorquí boast substantial census figures of 2119, 1037 and 654, respectively, indicating a strong interest and demand for these breeds’ hunting and ratting abilities.

Furthermore, the number of generations within each breed, calculated from the period that the studbook has worked until today, offers a glimpse into the breed’s evolution over time. Factors such as average breeding age and generation length contribute to determining the number of generations, highlighting the breed’s continuity and development across the years. For instance, assuming an average breeding age of 3 years and an average generation length of 3 years, approximately 7 to 8 generations, on average, have passed since the foundation of the studbook until today.

Aspects such as the breeds’ engagement in pet leisure activities, primary use (ranging from hunting/ratting to guarding/shepherding), and classification type (Breed Agroupation or Pure Breed) were evaluated. Numerical data are provided for various registry categories, including Foundational, Birth Auxiliary, Auxiliary, Definitive Birth, and Definitive registries, further segmented into females and males (Table 1). Additionally, total counts of animals across genders and registries within the studbook, owners, and breeders associated with each breed, were considered, providing a comprehensive foundation for a nuanced exploration of the intricate dynamics between indigenous dog breeds and their human companions in the Balearic Isles (Table 2).

### 2.2. Statistical Analysis

#### 2.2.1. A Priori Assumptions

Regularized Canonical correlation analysis (RCCA) assumes linearity, implying a linear relationship between the canonical variates and each set of variables. For RCCA to provide valid inferences, three key assumptions must be satisfied [14]. Firstly, like other multivariate test statistics, RCCA requires the variables to follow a multivariate normal distribution in the population, which is the multivariate counterpart of the bivariate normal distribution. Multivariate distributions may deviate from normality even if univariate or bivariate distributions are normal. It is important to note that with a large sample size (*n* > 1000), the Kolmogorov–Smirnov test becomes highly sensitive, and normality plots should be evaluated simultaneously. The Shapiro–Wilk test is recommended for assessing normality, especially for sample sizes exceeding 5000 [15]. In this study, normality was assessed using the Shapiro–Wilk test, revealing a non-normal distribution for both the different records in the studbooks as well as for the variables related to breeders and owners. Despite this, the D’Agostino–Pearson Test indicated that the data had been sampled from a normally distributed population (*p* > 0.05). It is worth mentioning that RCCA does not heavily rely on normality assumptions; instead, it emphasizes that the magnitude of coefficients in the correlation matrix should not be affected by large differences in variable distributions.

Secondly, homoscedasticity assumption needs to be checked [16]. Levene’s test was employed for this purpose, and as homoscedasticity was not met (*p* < 0.05), permutation tests were chosen for inference on canonical correlations, following Winkler’s recommendation [17].

Thirdly, a large sample size is required for canonical correlation analysis [16]. Some authors suggest a minimum of ten cases per variable, although this requirement may decrease as the sample size grows [18].

Multicollinearity and curvilinear relationships should also be considered. Multicollinearity, the phenomenon where one variable is almost a weighted average of others, and singularity, an exact relationship between variables, can impact RCCA results. Multicollinearity was assessed using the variance inflation factor (VIF), with a recommended maximum VIF of 5 [19]. In this study, the Multicollinearity statistics function of XLSTAT 2014 was used to compute VIF.

#### 2.2.2. Regularized Generalized Canonical Correlation Analysis (RCCA)

RCCA was conducted using the Canonical Correlation Analysis function in XLSTAT 2014.5.03 and SPSS MANOVA syntax. This analysis involves the regularization of covariance matrices for two sets of variables—a census per registry set comprising total, male, and female censuses per breed across registries in the studbook (Foundational Registry, Birth Auxiliary Registry, Auxiliary Registry, Definitive Birth, and Definitive Registry) of each breed (X) and an official members set which comprises the number of members that act as owners and/or breeders on each case (Y) by adding a multiple of the identity matrix (Id): Cov(X) + λ1Id and Cov(Z) + λ2Id. The regularization aims to reduce data dimensionality. Regularized Canonical Correlation Analysis was performed using the CCorA package in XLSTAT 2014.5.03. The CCorA package within XLSTAT facilitates the examination of linear associations between two sets of variables, offering insights into the underlying patterns and interdependencies. We estimated the Pearson product–moment correlation coefficient among variables from both sets using a bivariate procedure from the Correlate package of SPSS Statistics for Windows, Version 24.0, IBM Corp. [20] to avoid the severe multicollinearity or linear dependency between several variables, aiming to exclude those with multiple correlation coefficients higher than 0.80 according to Montgomery et al. [21].

#### 2.2.3. Validity

To establish the validity of RCCA, Pillai’s trace criterion was employed, which is a MANOVA test statistic ranging from 0 to 1. A higher Pillai’s trace indicates a stronger statistically significant linear relationship between the two variable groups. In this study, Pillai’s trace was highly significant (*p* < 0.01), confirming the validity of RCCA.

#### 2.2.4. Variability Explanation

Eigenvalues were calculated to determine the proportion of variance in canonical variates explained by canonical correlations. The largest eigenvalue corresponds to the first canonical correlation, and subsequent eigenvalues decrease in size. Canonical correlations, ranging from −1 to 1, were interpreted based on their magnitude. Canonical correlations ≥0.30 were considered meaningful, explaining about 10% of the variance.

Redundancy coefficients were used to assess how much variability in input variables was predicted by canonical variables. These coefficients indicate correlations between input variables and canonical variables, aiding in understanding their relationships.

Roots, representing the rank of eigenvalues, were used to test the null hypothesis that all correlations associated with roots were zero. Wilks’ lambda was employed to test the linkage of canonical variables to correlation tables.

#### 2.2.5. Canonical Correlation Analysis k-Fold Cross-Validation

Ten-fold cross-validation was performed to evaluate the validity and reduce sample-specific error. In this procedure, the sample was randomly divided into k subsamples, and the model was trained and validated on different subsamples. Cross-validation coefficients matched Wilk’s lambda values, confirming the validity of the analysis. Regularization parameters (λ1 and λ2) were selected via the *tune.rcc* function using a grid search, resulting in optimal values (λ1 = 0.001 and λ2 = 0.750). The choice of dimensions (d) was based on an empirical approach, considering a gap between the 1st and 2nd canonical correlations, leading to the inclusion of the first two dimensions in further analyses.

## 3. Results

Descriptive statistics for the two sets of variables—a census per registry set comprising total, male, and female censuses per breed across registries in the studbook (Foundational Registry, Birth Auxiliary Registry, Auxiliary Registry, Definitive Birth Registry, and Definitive Registry) of each breed, and an official members set which comprises the number of members that act as owners and/or breeders on each case—are provided in Table 2. Multicollinearity analysis indicated no variable should be discarded due to redundancies (VIF ≥ 5).

Pearson’s product–moment correlations were computed (Appendix A), revealing linear relationships among the census per registry set and the official members set. In general, weak to moderate correlations were observed.

Pillai’s trace criterion was highly significant (*p* < 0.01), indicating the validity of RCCA. The first canonical variate (F1) alone explained 50.00% of the variability in both datasets.

A canonical correlation of 1, indicative of a perfect correlation between the two sets of variables, was found. This implies not only the fact that as number of owners and breeders increases, the number of dogs across registries does, too, but also that the same tone is followed, meaning each owner breeder may likely contribute to the registration of just one animal. Redundancy coefficients measure the proportion of variance in one set of variables that can be explained by the other set of variables. In this case, a redundancy coefficient of 0.637 for function 1 and 0.363 for function 2 indicates that 63.7% of the variance in function 1 can be explained by the other set of variables, while 36.3% of the variance in function 2 can be explained by the other set of variables.

In canonical correlation analysis, standardized canonical coefficients and loadings are used to measure the strength and direction of the relationship between the variables in each set. Standardized canonical coefficients indicate the extent to which each variable contributes to the latent variable. Standardized canonical loadings indicate the strength and directionality of the relationship between the measured variable and the latent variable. Standardized canonical loadings are reported in Figure 1, while standardized canonical coefficients are reported in Figure 2.

Wilks’ lambda and R2 results indicated the significance of all variates, and the multivariate generalization of R2 highlighted factors contributing to the explanatory potential of the Census per registry and Official members data.

A ten-fold cross-validation confirmed the validity of RCCA. Optimal regularization parameters were selected, and the analysis included the first two dimensions, explaining 100.00% of the variance.

Figure 3 depicts the results of the ten-fold cross-validation, with coefficients matching Wilk’s lambda values. Optimal regularization parameters (λ1 = 0.001 and λ2 = 0.750) were selected through a grid search (Figure 3).

Ten-fold cross-validation is a pivotal technique for rigorously assessing the performance of models. This method involves dividing the dataset into ten equal parts or folds. Through a systematic process, the model is trained on nine folds while the performance is evaluated on the remaining fold. This cycle is repeated ten times, ensuring each fold serves as a validation set at least once. By averaging the performance metrics across all iterations, ten-fold cross-validation provides a reliable measure of how well the model generalizes to unseen data.

In statistical analysis, Wilk’s lambda emerges as a significant measure, particularly in multivariate analysis of variance (MANOVA) or linear discriminant analysis (LDA). It quantifies the extent to which independent variables explain variance in dependent variables. Coefficients matching Wilk’s lambda values become crucial, especially when employing regularization techniques like L1 or L2 regularization. Adjusting the model’s coefficients to optimize the explained variance while adhering to regularization constraints becomes paramount in this context.

Moreover, regularization plays a pivotal role in preventing overfitting by introducing penalty terms into the model’s loss function. The regularization parameters, denoted as λ1 and λ2 for L1 (Lasso) and L2 (Ridge) regularization, respectively, are essential hyperparameters to be tuned. Through grid search techniques, various combinations of these parameters are explored to identify the optimal values. In this case, the optimal regularization parameters are determined to be λ1 = 0.001 and λ2 = 0.750, striking a balance between model complexity and performance enhancement [22].

The study concluded that the RCCA provided a valid analysis of the relationship between different studbook records and variables related to breeders and owners. The regularization process reduced dimensionality, and the first two dimensions explained a substantial portion of the variance.

## 4. Discussion

Considering the aforementioned framework, the discussion of the correlations within the first variable group of records reveals significant patterns. An increase in the auxiliary registration of females corresponds to a twofold decrease in the foundational registration of females. This phenomenon is attributed to the closure of foundational registrations and the opening of auxiliary registrations, allowing for the inclusion of animals in genealogical records without known pedigrees, particularly noticeable in male registrations. As registrations commence, there is a period during which foundational registrations remain open for both male and female specimens that have undergone breed confirmation. Subsequently, foundational registrations close, and the entries in this registry persist throughout the animal’s lifetime [23]. Moreover, an increase in the definitive registrations of females results in a twofold decrease in the foundational registrations of females, correlating with a heightened awareness of pedigrees in subsequent generations. Historical disinterest in pedigrees, with scant details on paternity, shifted as interest in dog shows and accurate registration details grew, leading to the development of genealogical records with detailed ancestral information [24].

Additionally, a noteworthy correlation exists within the first variable: as the initial registration of females rises, the total definitive registration and overall animal count increase at a ratio of 1:0.533 and 1:0.957, respectively. This suggests the importance of a robust initial registration of females in building a substantial number of animals with detailed genealogical information. Jerold S. Bell’s conference on [25] “Inbreeding, Consanguinity, and the Evolution of Breeds” in 2013 underscores that the genealogical record in a breed’s early stages may include individuals of unknown ancestry or those conforming to the breed’s conformational or working standards. These individuals serve as the foundational stock of the breed, and as the genealogical record continues, knowledge of pedigrees increases.

Moreover, the study observes an intriguing correlation during the period when foundational registrations were open, indicating a preference for females. Conversely, Diverio et al. [26] reveal in their article that most Italian men prefer intact male dogs. As the auxiliary registration of females increases, a proportional rise occurs in the definitive registration of females, as well as in the auxiliary and definitive registrations of males. This suggests that higher female registrations, regardless of the category, lead to registrations with more extensive pedigree knowledge, particularly in definitive registrations. The study findings align with the observations of Alanzor Puente, Pons Barro et al. [27], noting that, due to generational intervals, male dogs with guarding functionality are preferred over females.

Shifting focus to the second variable of owners and breeders, a weak correlation is evident. The larger the number of breeders and owners, the larger the number of all initial registrations, the total definitive registration, and the overall animal count, with breeders exhibiting nearly double the impact compared to owners. The number of breeders has been reported as an indirect indicator of breed popularity, which, in turn, leads to the increase in the number of births and, thus, puppies, which are eventually registered, and owners that acquire such puppies [28].

This breeder/owner number discrepancy may be explained by the active involvement of breeders in the early stages of breeding programs when foundational registrations are open. Here, breeders, predominantly forming breed associations, play a prominent role in genealogical record-keeping, being the first to register their specimens. The proportion of male and female registrations slightly favors males for both owners and breeders, with a doubling in favor of breeders. A decrease in breeders leads to a half increase in all auxiliary and definitive registrations, a trend not observed with owners. This variation could be attributed to differences in interest regarding genealogical knowledge between breeders and owners.

The selling/buying transactions of puppies in the particular case of endangered dog breeds in the Balearic Islands is minoritary, and, generally, newborn puppies act as a source for owners or breeders who are already members of breeders’ associations. This occurs with the exception of specific cases, such as Ca de Rater and Ca de Bestiar [27,29], for which external sells of exceptional puppy occur, and Ca de Bou, whose puppies have anecdotally been sold internationally (for instance, in Finland, Norway, and Germany). This internal breeder and owner connection enables the protection of genetic diversity, but also compels us to apply strategies that seek the minimization of related individuals ending up with breeders who would eventually mate them, increasing inbreeding and co-ancestry levels.

Analyzing the correlation between the first (F1) and second (F2) discriminant functions (Figure 1 and Figure 2) reveals a balanced proportion of 1:1. Consequently, an increase in the number of breeders and owners results in a proportional rise in the number of animals across different registries. The analysis underscores that each function or variable explains 50% of the variability among functions, different genealogical record categories, and owners–breeders dynamics.

In the broad context of previous studies, these findings enrich our understanding of the dynamics within genealogical records and the roles of breeders, owners, and different registration types. The observed correlations align with historical shifts in the interest and practices surrounding genealogical records. The closure of foundational registrations and the subsequent rise in auxiliary registrations illuminate the evolutionary nature of record-keeping practices, reflecting changes in breeding strategies and the increasing importance placed on documented pedigrees.

The significant correlation between the increase in the definitive registrations of females and the decline in foundational registrations suggests an evolving emphasis on pedigree knowledge over generations. This mirrors broader trends in dog breeding communities, where a growing appreciation for accurate pedigree information coincides with the rise of dog shows and heightened interest in breed standards. The study’s findings contribute valuable insights into the transformation of attitudes towards genealogical documentation.

Furthermore, the proportional relationship between the initial registration of females and the subsequent surge in total definitive registrations and overall animal counts underscores the enduring impact of foundational records. This emphasizes the enduring legacy of well-documented initial registrations, indicating that a robust start in record-keeping can lead to a wealth of detailed genealogical information over time. Such insights, derived from the correlation analysis, carry implications for breed conservation and management strategies, emphasizing the importance of meticulous initial registrations.

The observed preferences for female registrations during the foundational registration period align with historical practices, providing a nuanced understanding of gender preferences among breeders and owners. The proportional increase in definitive registrations, both for females and males, as the auxiliary registration of females rises, hints at a collective effort to enhance genealogical knowledge. This finding suggests that regardless of gender preferences during initial registrations, the focus shifts towards comprehensive pedigree information at subsequent registration stages. In other words, while there may be a preference for registering females initially, the focus shifts towards ensuring thorough genealogical information for both genders as the registration process progresses and more animals are registered in the definitive registry. This suggests a broader commitment to maintaining accurate and detailed records of purebred dogs’ pedigrees regardless of gender biases during the initial registration phase [1]. Indeed, over the past few centuries, kennel clubs have gathered samples of working and hunting breeds, which were isolating them sexually, thus creating, at best, historical representations of the original working or hunting breeds, as suggested by Lord et al. [30].

Turning to the second variable, the correlation between breeders and owners and their impact on registrations provides valuable insights into the early stages of breeding programs. The dominance of breeders in influencing initial registrations highlights their pivotal role in shaping genealogical records. The observed gender preferences among breeders and owners further contribute to our understanding of the intricate dynamics within breeding communities.

An essential finding emerges concerning the impact of female auxiliary registrations on foundational registrations, signaling a transformative shift in breed documentation practices. The diversification of studbook registries into foundational, birth, auxiliary, and definitive categories holds paramount importance in the comprehensive documentation and preservation of dog breeds. The foundational registry captures the initial entries during a breed’s inception, serving as the bedrock for genetic diversity. Birth registries mark the introduction of new generations, providing a dynamic snapshot of a breed’s evolution. Auxiliary registries play a crucial role by accommodating animals with unknown genealogy, ensuring flexibility in incorporating diverse lineages. The definitive registry, on the other hand, signifies a meticulous confirmation of a dog’s pedigree, offering a reliable repository of ancestral information. This stratification allows for a nuanced understanding of breed dynamics, genetic lineage, and historical shifts in preferences, aiding breeders, researchers, and enthusiasts alike. By recognizing the distinct roles of each registry type, studbooks contribute significantly to the preservation of genetic heritage and the sustainable management of dog breeds over time. These findings offer a nuanced perspective on the evolution of genealogical record-keeping practices and the roles of breeders and owners in shaping these records. The implications extend to broader discussions on breed conservation, breeding program management, and the evolving attitudes towards genealogical documentation [1,29,31,32,33]. Future research directions may explore the specific factors influencing the observed correlations, delve deeper into the motivations of breeders and owners, and investigate the broader societal trends impacting genealogical practices in dog breeding. The intricate interplay between historical practices and contemporary shifts in attitudes towards genealogical records remains a rich area for further exploration.

## 5. Conclusions

In summary, the connection between the records of Balearic dog breeds and their owners, whether breeders or owners, highlights the crucial role of these records in maintaining a well-functioning pedigree book and understanding the lineage of registered specimens. The establishment of a strong foundation in pedigree books necessitates comprehensive initial records, ideally supported by molecular-marker-assisted breed genetic adscription techniques, adhering to breed standards. As the foundational record is closed and pedigree book regulations are met, final records increase while initial ones decrease. Females in initial records are emphasized for their significant role in expanding the breed and ensuring a greater number of individuals for future generations, particularly important for endangered breeds. This study emphasizes the essential contributions of breeders and owners in conserving genetic resources, with breeders having a more prominent role in foundational records. There is a noted shift in gender ratios from more females in initial records to a slight trend favoring males in the final records, a choice influenced by both breeders and owners. The study identifies a gap regarding unreported losses of specimens in genealogical records, urging future research to address this issue for the association to maintain reliable and updated censuses.

## Figures and Tables

**Figure 1 animals-14-00639-f001:**
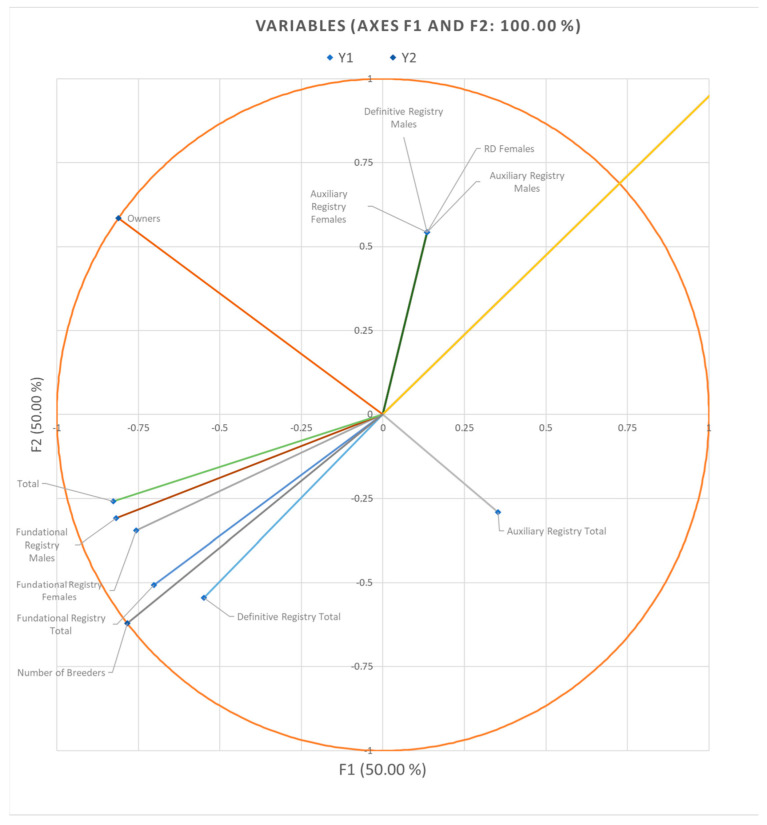
Standardized canonical loadings for the census per registry set and the official members set.

**Figure 2 animals-14-00639-f002:**
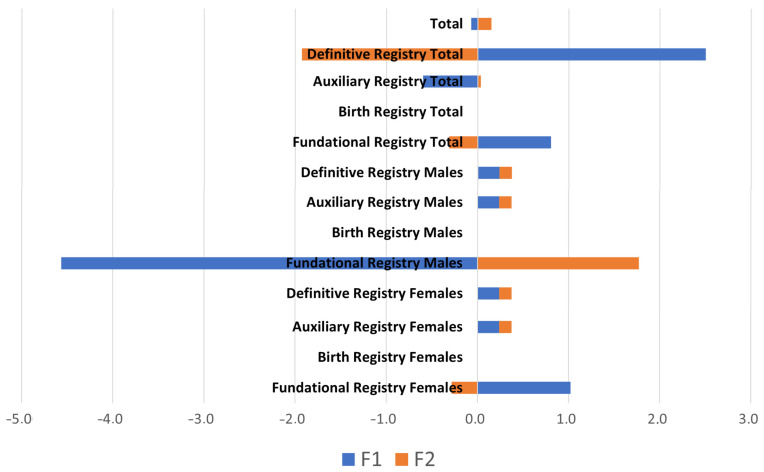
Standardized canonical coefficients for the census per registry set and the official members set.

**Figure 3 animals-14-00639-f003:**
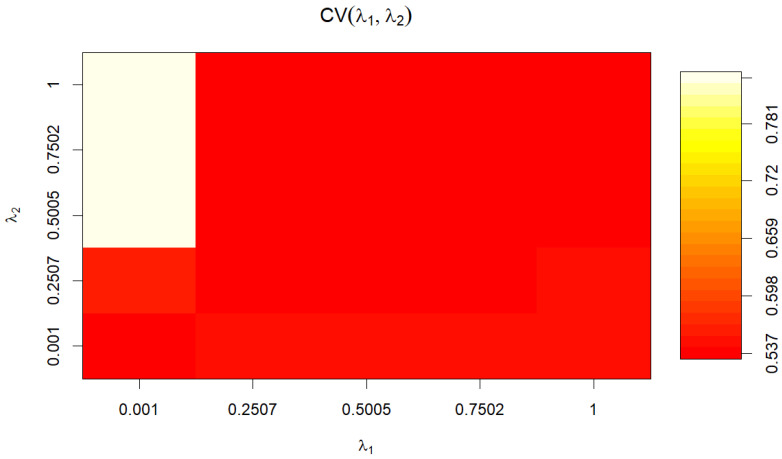
Optimal regularization parameters (λ1 = 0.001 and λ2 = 0.750) selected through grid search.

**Table 1 animals-14-00639-t001:** Classification chart for registry type, description, similarities and differences across registries considered within the studbook of canine breeds.

Registry Type	Description	Similarities	Differences
Foundational (FR)	This is the Foundational Registry. It involves the initial registration of a new breed or variety of dogs. The RF includes the documentation of the breed’s origins, lineage, and essential characteristics. It serves as the starting point for establishing a new breed within the official registry system.	Establishes the base for breed standards.	Limited information compared to other registries.
Definitive (DR)	The Definitive Registry comprises dogs that meet all the breed standards and have undergone thorough evaluation. These dogs are officially recognized as purebred representatives of their breed. The RD ensures that the breed adheres to specific criteria, such as appearance, temperament, and genetic heritage.	Contains complete lineage details.	Most accurate and detailed registry type.
Auxiliary (AR)	The Auxiliary Registry includes dogs that do not fully meet the criteria for the RD but still have valuable genetic contributions. These dogs may be used for breeding purposes, even though they might not exhibit all the desired breed traits. The AR helps maintain genetic diversity within the breed.	Supplements birth records.	May still lack complete lineage details.
Birth Definitive (BDR)	The BDR confirms birth details and establishes lineage. It adds certainty to lineage records by verifying birth information and ensuring accurate documentation of parentage.	Adds certainty to lineage.	May still lack comprehensive breed history.
Birth Auxiliary (BAR)	The Birth Auxiliary Registry records the birth of individual puppies within a breed. It documents their parentage, birth date, and other relevant details. The BAR is essential for maintaining accurate lineage records and tracking the breed’s population.	Tracks birth details.	May not include full lineage information.
Merit (MR) (Currently not defined for any of the canine breeds in the Balearic Islands)	The MR recognizes exceptional dogs within a breed. These dogs have achieved specific accomplishments, such as winning championships, excelling in performance events, or contributing significantly to the breed’s improvement. The MR celebrates excellence and encourages breed development.	Recognizes exceptional dogs.	Focuses on accomplishments rather than birth or lineage.

**Table 2 animals-14-00639-t002:** Descriptive statistics for the census per registry set comprising total, male and female censuses per breed across registries in the studbook (Foundational Registry, Birth Registry, Auxiliary Registry, Risk Priority Number, and Definitive Registry) of each breed, and an official members set which comprises the number of members that act as owners and/or breeders in each case.

Set	Gender	Variable	Minimum	Maximum	Mean	Std. Deviation
Census per registry	Females	Foundational Registry	129	1114	463	401.998
		Birth Auxiliary Registry	23	23	23	0 ^a^
		Auxiliary Registry	60	98	79	12.017
		Birth Definitive Registry	2	2	2	0 ^a^
		Definitive Registry	19	162	90.500	45.221
	Males	Foundational Registry	96	1005	298.333	353.799
		Birth Auxiliary Registry	7	7	7	0 ^a^
		Auxiliary Registry	5	93	49	27.828
		Birth Definitive Registry	0	0	0	0 ^a^
		Definitive Registry	1	154	77.500	48.383
	Total	Foundational Registry	243	2119	850.400	679.306
		Birth Auxiliary Registry Total	30	30	30	0 ^a^
		Auxiliary Registry	65	316	190.667	79.373
		Birth Definitive Registry	2	2	2	0 ^a^
		Definitive Registry	20	2119	777.250	721.341
		Total	243	2119	812.667	699.519
Official members	Number	Owners	38	359	225.333	132.539
		Breeders	20	86	50.600	21.814

^a^ The distinction between auxiliary and definitive birth registries is not made by all Balearic dog breed associations in their studbooks, comprising instead a single birth registry. In these cases, the results obtained across the breeds that account for this birth registry subdivision were the same, thus a value of zero is reported for SD.

## Data Availability

Data are contained within the article.

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
