# Peer review of "The Crucial Role of Breeder and Dog Owners Associations in Safeguarding Genetic Heritage of Endangered Balearic Dog Breeds: Gender Preference and Registry Adscription"

_animals, 2024, doi:10.3390/ani14040639_

Round 1

Reviewer 1 Report

Comments and Suggestions for Authors

Line 19 and line 33: The type of registries need definition in some point of the text, maybe not in the Summary or the abstract, but it seems necessary to better understand the different types of registries, one of the core information’s of the article.

Line 48: the breeds should be listed here, it is more appealing to continue to read the article if the reader has a clear idea about the subject.

Line 57: Additionally…should be in the objectives or in the results.

Line 97: Bring the information of the type of dogs to the introduction.

Line 132: large sample? – be more precise

Line 147: Birth and

Table 1: why are there no variations in the Birth auxiliary registry?

4. Discussion

It will be interesting to know if registration of the stud animals is mandatory for reproduction.

Line 273: delete in the article

Line 282 – 286: It is difficult to understand the pertinence of the information. Why breeders and owners increase… the breed is more popular… more prolific, therefore are more puppies, more owners…

Line 294: first and second functions- not clear

Again, it will be ideal to know the requirement of breeding these types of dog; and to take them in dogs’ expositions.

Line 307: Are registrations important for increasing the price of the puppies?

Line 324-326: How did you get to this conclusion?

Line 344: Delete in the summary. “underscores the connection” meaning

Line 348: breed assignment techniques: define

Comments on the Quality of English Language

The english needs minor reviewing. However, some of the information presented in the discussion are not clear.

Author Response

Reviewer: 1

Line 19 and line 33: The type of registries need definition in some point of the text, maybe not in the Summary or the abstract, but it seems necessary to better understand the different types of registries, one of the core information’s of the article.

Response: We added a chart to provide a clear description of the registries considered in the present study.

Line 48: the breeds should be listed here, it is more appealing to continue to read the article if the reader has a clear idea about the subject.

Response: Added.

Line 57: Additionally…should be in the objectives or in the results.

Response: Suggestion was followed.

Line 97: Bring the information of the type of dogs to the introduction.

Response: Sugegstion was followed.

Line 132: large sample? – be more precise

Response: As stated afterwards (n>1000) samples over 1000 observations.

Line 147: Birth and

Response: Corrected.

Table 1: why are there no variations in the Birth auxiliary registry?

Response: We clarifies this in Table 1 (now Table 2). This derives from two facts. The distinction between auxiliary and definitive birth registries is not made by all Balearic dog breed associations in their studbooks, comprising isntead a single birth registry. In these cases, the results obtained across the breeds that account for this birth registry subdivision were the same, thus 0 value reported for SD.

  1. Discussion

It will be interesting to know if registration of the stud animals is mandatory for reproduction.

Response: Reviewer suggestion was included in the introduction as we feel it may fit better.

Line 273: delete in the article

Response: Deleted.

Line 282 – 286: It is difficult to understand the pertinence of the information. Why breeders and owners increase… the breed is more popular… more prolific, therefore are more puppies, more owners…

Response: This was clarified in the body text.

Line 294: first and second functions- not clear.

Response: We clarified this in the body text.

Again, it will be ideal to know the requirement of breeding these types of dog; and to take them in dogs’ expositions.

Response: We clarified this at the beginning of the discussion.

Line 307: Are registrations important for increasing the price of the puppies?

Response: Selling/buying transactions of puppies in the particular case of endangered dog breeds in the Balearic Islands is minoritary and generally newborn puppies act a source for owners or breeders who are already members of breeders associations. This occurs with the exception of specific cases, such as Ca de Rater and Ca de Bestiar, for which exceptional puppy external sells occur and Ca de Bou, whose puppies have anecdotally been sold internationally (for instance, Finland, Norway and Germany).

Line 324-326: How did you get to this conclusion?

Response: We clarified it in the body text.

Line 344: Delete in the summary. “underscores the connection” meaning

Response: We deleted it following reviewer suggestion.

Line 348: breed assignment techniques: define

Response: We clarified it.

Comments on the Quality of English Language

The english needs minor reviewing. However, some of the information presented in the discussion are not clear.

Response: We thank the reviewer for his/her kind comments. A Cambridge University ESOL examinantion instructor went through the manuscript to correct potential grammar mistakes or typos which could compromise reading and understanding.

Reviewer 2 Report

Comments and Suggestions for Authors

The paper offers a novel approach to historical and geographical analyses in dog breeding. As such the authors are to be lauded for this idea.

However I suggest a substantial re-organization and extension of the ms.

But first, a general request: Please avoid if possible the term "owner": No sentient being should be the property of someone else.

Apart from that,  we need more info

-  on the breeds: What is their original purpose? what is the sample size PER BREED? what time span / how many generations did the analyses cover?

The intro section should be re-organized. Sections from line 55 to 79 are parts of the conclusions and thus should be transferred towards the end of the discussion.

Some more literature also shouldf be added, e.g. the work by the Coppinger group an working dogs, or the new publication by Chira et al 2024 (Scient Rspecially fig 3 needs some explanation....eports) on the relationship between previous usage of dogs  and the way people think about them...

At the end of the results section we would profit from some explanatory sentences - what does all this mean??

Especially fig 3 needs explanation!!

Author Response

The paper offers a novel approach to historical and geographical analyses in dog breeding. As such the authors are to be lauded for this idea.

Response: We thank the reviewer for his/her kind comments.

However I suggest a substantial re-organization and extension of the ms.

Response: suggestions by the reviewer were followed.

But first, a general request: Please avoid if possible the term "owner": No sentient being should be the property of someone else.

Response: We considered changing the term to caretaker, but administratively they are considered owners, and sometimes these caretaker and owner differs. Hence, we feel it would be challenging. Although we understand and agree with reviewer.

Apart from that,  we need more info

-  on the breeds: What is their original purpose? what is the sample size PER BREED? what time span / how many generations did the analyses cover?

Response: We followed the reviewer suggestion.

The intro section should be re-organized. Sections from line 55 to 79 are parts of the conclusions and thus should be transferred towards the end of the discussion.

Response: reviewer suggestion was followed.

Some more literature also shouldf be added, e.g. the work by the Coppinger group an working dogs, or the new publication by Chira et al 2024 (specially fig 3 needs some explanation....) on the relationship between previous usage of dogs  and the way people think about them...

Response: We followed the reviewer sugegstion.

At the end of the results section we would profit from some explanatory sentences - what does all this mean??

Especially fig 3 needs explanation!!

Response: We followed the reviewer’s suggestion.